# Associations between Early Surgery and Postoperative Outcomes in Elderly Patients with Distal Femur Fracture: A Retrospective Cohort Study

**DOI:** 10.3390/jcm10245800

**Published:** 2021-12-11

**Authors:** Norio Yamamoto, Hiroyuki Ohbe, Yosuke Tomita, Takashi Yorifuji, Mikio Nakajima, Yusuke Sasabuchi, Yuki Miyamoto, Hiroki Matsui, Tomoyuki Noda, Hideo Yasunaga

**Affiliations:** 1Department of Orthopedic Surgery, Miyamoto Orthopedic Hospital, Okayama 773-8236, Japan; norio-yamamoto@umin.ac.jp; 2Department of Epidemiology, Graduate School of Medicine, Dentistry and Pharmaceutical Sciences, Okayama University, Okayama 700-8558, Japan; yorichan@md.okayama-u.ac.jp; 3Systematic Review Workshop Peer Support Group (SRWS-PSG), Osaka 541-0043, Japan; 4Department of Clinical Epidemiology and Health Economics, School of Public Health, The University of Tokyo, Tokyo 113-0033, Japan; mikioh@ks.kyorin-u.ac.jp (M.N.); mimonism@yahoo.co.jp (Y.M.); ptmatsui-tky@umin.ac.jp (H.M.); yasunagah-tky@umin.ac.jp (H.Y.); 5Department of Physical Therapy, Faculty of Health Care, Takasaki University of Health and Welfare, Takasaki 370-0033, Japan; tomita-y@takasaki-u.ac.jp; 6Emergency Life-Saving Technique Academy of Tokyo, Foundation for Ambulance Service Development, Tokyo 192-0364, Japan; 7Data Science Center, Jichi Medical University, Tochigi 329-0498, Japan; rhapsody77777@gmail.com; 8Department of Emergency Medicine, Kyoto Prefectural University of Medicine, Kyoto 602-8566, Japan; 9Department of Orthopaedic Surgery and Traumatology, Kawasaki Medical School General Medical Center, Okayama 700-8505, Japan; tnoda@med.kawasaki-m.ac.jp

**Keywords:** distal femur fracture, surgical timing, mortality, complications, length of hospital stay, medical costs, database

## Abstract

Previous literature has provided conflicting results regarding the associations between early surgery and postoperative outcomes in elderly patients with distal femur fractures. Using data from the Japanese Diagnosis Procedure Combination inpatient database from April 2014 to March 2019, we identified elderly patients who underwent surgery for distal femur fracture within two days of hospital admission (early surgery group) or at three or more days after hospital admission (delayed surgery group). Of 9678 eligible patients, 1384 (14.3%) were assigned to the early surgery group. One-to-one propensity score matched analyses showed no significant difference in 30-day mortality between the early and delayed groups (0.5% versus 0.5%; risk difference, 0.0%; 95% confidence interval, −0.7% to 0.7%). Patients in the early surgery group had significantly lower proportions of the composite outcome (death or postoperative complications), shorter hospital stays, and lower total hospitalization costs than patients in the delayed surgery group. Our results showed that early surgery within two days of hospital admission for geriatric distal femur fracture was not associated with a reduction in 30-day mortality but was associated with reductions in postoperative complications and total hospitalization costs.

## 1. Introduction

The incidence of distal femur fracture increases markedly with age [1]. Elderly patients with distal femur fracture were reported to have similar demographic characteristics and outcomes to elderly patients with hip fracture [2]. Elderly patients with distal femur fracture have poor postoperative outcomes because of their many perioperative complications caused by preoperative immobilization [2,3], which is similar to the case for elderly patients with hip fracture [4]. In elderly patients with hip fracture, systematic reviews have demonstrated the beneficial effects of early surgery on postoperative outcomes, including mortality and postoperative complications [5,6].

However, prior studies have provided mixed results regarding the association between the timing of surgery and postoperative mortality in elderly patients with distal femur fracture. Several observational studies found that early surgery was associated with reduced mortality [7,8,9], while other studies showed no such association [10,11,12,13]. One of the reasons for these conflicting results may be small patient numbers used in previous studies (*n* = 88 to 392) [2,3,7,8,9,10,11,12]. In addition, confounding factors may not have been sufficiently adjusted for because of the small numbers of patients.

The purpose of the present study was to examine the associations between early surgery and postoperative outcomes in elderly patients with distal femur fractures using data from a nationwide inpatient database. We hypothesized that early surgery is associated with a lower in-hospital mortality, a lower proportion of postoperative complications, and lower medical costs. By clarifying these associations, we suggest a better treatment strategy for geriatric distal femur fracture that will improve patient outcomes and the social economy of healthcare.

## 2. Materials and Methods

This study was a retrospective cohort study using a national administrative inpatient database under the REporting of studies Conducted using Observational Routinely-collected Data (RECORD) statement reporting guidelines [14]. This study was conducted in accordance with the Declaration of Helsinki and was approved by the Institutional Review Board of The University of Tokyo (approval number: 3501-(3); 25 December 2017). The requirement for informed consent was waived because of the anonymous nature of the data.

### 2.1. Data Source

We used the Japanese Diagnosis Procedure Combination inpatient database under the management of the Ministry of Health, Labour, and Welfare, which includes administrative claims data and discharge abstracts from more than 1600 acute-care hospitals and covers approximately 90% of all tertiary emergency hospitals in Japan [15]. The database includes information on age, sex, body weight, body height, smoking history, level of consciousness at admission, home medical care use, location before hospital admission, ambulance use, diagnoses recorded with International Classification of Diseases Tenth Revision (ICD-10) codes, treatments recorded with Japanese medical procedure codes, medications administered, discharge status, and hospitalization costs [15]. The diagnoses at admission, comorbidities at admission, and complications during hospitalization are recorded in the database. The attending physicians are required to report objective evidence for their diagnoses for the purpose of cost reimbursement because the diagnostic records are linked to a payment system [15]. In a validation study of the database, the recorded procedures had a high sensitivity and specificity, while the recorded diagnoses had a high specificity and moderate sensitivity [16]. We also used facility information and statistics data from the Survey of Medical Institutions 2015 [17]. We combined these data with the data from the Japanese Diagnosis Procedure Combination inpatient database using specific hospital identifiers. The Survey of Medical Institutions data included the hospital type (academic hospital, teaching hospital, or tertiary emergency hospital) and the number of hospital beds.

### 2.2. Patient Selection

We searched the database and included patients who: (i) were admitted for distal femur fracture (ICD-10 code, S724), (ii) underwent surgery for distal femur fracture during hospitalization, and (iii) were discharged between April 2014 and March 2019. We excluded patients who: (i) were less than 60 years of age, (ii) were transferred from another hospital, (iii) had subsequent admission for distal femur fracture during the study period, (iv) were admitted to hospitals that could not be linked with data from the Survey of Medical Institutions 2015, (v) were admitted for open distal femur fracture (ICD-10 code, S7241), (vi) were treated with external fixation, (vii) had non-union, (viii) had bone tumor, and (ix) underwent surgery beyond 12 days after hospital admission.

### 2.3. Main Exposure

The main exposure was timing to surgery after hospital admission. We divided the eligible patients into an early surgery group (surgery within two days of hospital admission) and a delayed surgery group (surgery at three or more days after hospital admission). We defined the timing for early surgery as within two days of hospital admission, because this was considered representative timing for surgery in previous studies on distal femur fracture and hip fracture [7,8,10,18].

### 2.4. Outcomes

The primary outcome was the all-cause 30-day in-hospital mortality. Patients discharged alive within 30 days of hospital admission were considered alive at 30 days. The 30-day observation period was the common follow-up period used in previous studies [2,19]. The secondary outcomes were all-cause in-hospital mortality, composite outcome of death or postoperative complications during hospitalization, length of hospital stay, length of time from surgery to discharge, and total hospitalization costs [19]. The composite outcome was defined as death during hospitalization or at least one postoperative complication in the post-admission complication diagnoses detected by relevant ICD-10 codes (listed in Appendix A) [20]. Length of hospital stay was defined as the duration of hospital admission to hospital discharge. The exchange rate for total hospitalization costs was set at 1 US dollar to 110 Japanese yen.

### 2.5. Covariates

The covariates were age; sex; body mass index at admission; smoking history; level of consciousness at admission using the Japan Coma Scale, which is well correlated with the Glasgow Coma Scale [21]; home medical care use; admission from nursing home; ambulance use, admission on a weekend (Friday to Sunday); calendar year; comorbidities according to the ICD-10 codes (listed in Appendix A), Charlson comorbidity index [22]; ICD-10-based trauma mortality prediction score [23]; intensive care unit or high care unit use at admission; fracture type of periprosthetic fracture; operative method used for distal femur fracture; and hospital characteristics. We comprehensively selected these covariates as confounders based on existing literature with clinical judgment [18,19].

Body mass index (kg/m^2^) was categorized as <18.5, 18.5–24.9, 25.0–29.9, ≥30.0, and missing data. An ICD-10-based trauma mortality prediction score was developed and validated by Wada et al. [23]; it has achieved a high accuracy for mortality prediction in the Japanese Diagnosis Procedure Combination inpatient database. Patients were considered to have severe medical conditions at admission if they were admitted to an intensive care unit or a high care unit on the day of admission [18]. We defined hospital volume as the number of operations for distal femur fracture during the study period.

### 2.6. Statistical Analysis

A propensity score matching method was applied to compare the outcomes between the early and delayed surgery groups. First, we performed a multivariable logistic regression analysis to estimate the propensity scores for patients receiving early surgery using all covariates listed in Table 1. Briefly, we performed one-to-one nearest-neighbor matching without replacement using a caliper width set at 20% of the standard deviation for the estimated propensity scores [24]. Absolute standardized differences were calculated for all covariates in the unmatched and matched cohorts to confirm the balance of the covariate distributions between the early and delayed surgery groups. The imbalance in the covariate distributions was considered negligible when the absolute standardized differences between the two groups were less than 10% [24]. The propensity score matching was performed using the PSMATCH2 module of STATA (Edwin Leuven and Barbara Sianesi) [25]. We calculated risk differences and their 95% confidence intervals for the outcomes using a generalized linear model with the identity link function and with cluster-robust standard errors for individual hospitals as clusters.

Subsequently, we performed the following four subgroup analyses to estimate the heterogeneity of the treatment effect on the primary outcome in the propensity score-matched cohort: patients with admission on the weekend, patients with a Charlson comorbidity index ≥1, patients with a hospital size < 400 beds, and patients with a hospital volume <4.

Categorical variables were presented as numbers and percentages, while continuous variables were presented as means and standard deviations (SDs). All *p*-values were two-sided and values of *p* < 0.05 were considered statistically significant. All analyses were performed using the STATA/MP 16.0 software (StataCorp, College Station, TX, USA).

## 3. Results

After the application of the exclusion criteria, a total of 9678 patients were eligible for the study (Figure 1). The mean (SD) age was 81.1 (9.3) years, and 90.3% were women. Among the eligible patients, 1384 (14.3%) were assigned to the early surgery group and 8294 (85.7%) were assigned to the delayed surgery group. Surgery on day 4 after hospital admission (15.4%) was the most common timing (Figure 2). The mean (SD) timing for surgery after hospital admission was 1.8 (0.4) days in the early surgery group and 5.9 (2.3) days in the delayed surgery group.

Table 1 shows the patient characteristics in the two groups before and after propensity score matching. Before propensity score matching, patients in the early surgery group tended to have intensive care unit or high care unit use at admission and be admitted to a hospital with a high hospital volume, while patients in the delayed surgery group tended to have admission on the weekend, comorbidities of rheumatic diseases and renal dysfunction, and a higher Charlson comorbidity index. The proportions of periprosthetic fracture were 6.6% in the early surgery group and 8.1% in the delayed surgery group. After propensity score matching, the patient characteristics were found to be well-balanced between the two groups (Appendix A). The propensity score distributions before and after propensity score matching are shown in Appendix A.

Table 2 shows the outcomes before and after propensity score matching. Before propensity score matching, 30-day mortality in the early and delayed surgery groups was 0.9% and 0.5%, respectively. After propensity score matching, there was no significant difference in 30-day mortality between the two groups (risk difference, 0.0%; 95% CI, −0.7% to 0.7%). There was no significant difference in in-hospital mortality between the two groups. The composite outcome was significantly less common in the early surgery group compared with the delayed surgery group (13.5% versus 17.1%; risk difference, −3.7%; 95% CI, −6.5% to −0.9%). Regarding postoperative complications, acute coronary syndrome was significantly less frequent in the early surgery group compared with the delayed surgery group (Appendix A). Patients in the early surgery group had a significantly shorter length of hospital stay (risk difference, −8.4 days; 95% CI, −11.8 to −5.0 days), a shorter length of time from surgery to discharge (risk difference, −4.5 days; 95% CI, −7.9 to −1.0 days), and lower total hospitalization costs (risk difference, −2101 US dollars; 95% CI, −2991 to −1212 US dollars) than patients in the delayed surgery group.

The four subgroup analyses showed no significant differences in 30-day mortality between the two groups and the results were consistent with those in the main analysis (Table 3).

## 4. Discussion

The present study was a large-scale investigation of the effectiveness of early surgery for distal femur fracture in elderly patients using data from a nationwide inpatient database in Japan. The results showed that early surgery within two days of hospital admission was not associated with a reduction in 30-day mortality but was significantly associated with reductions in postoperative complications, length of hospital stay, and total hospitalization costs.

Early surgery did not have clinical effectiveness in reducing 30-day mortality. This finding was consistent with that of previous studies on geriatric distal femur fracture that assessed outcomes with covariate adjustment using a multivariable logistic regression model [10,12]. However, the finding that early surgery in elderly patients with distal femur fracture did not reduce 30-day mortality differed from results gained in elderly patients with hip fracture [19].

Regardless of the timing of surgery, the 30-day mortality in the present study was less than 1% and much lower than the rates of 3% to 8% seen in previous studies on geriatric distal femur fracture [2,7,10,11]. However, the low 30-day mortality was consistent with the 30-day mortality rates in geriatric hip fracture studies that used data from large databases (0.6% to 1.0% in Japan; 5.8% to 6.5% in Canada; 6.3% in Sweden) [3,18,19,20]. In our cohort with a very low 30-day mortality, early surgery did not have clinical effectiveness for 30-day mortality.

Early surgery was associated with better clinical outcomes and lower costs, as demonstrated by the lower frequency of the composite outcome, shorter length of hospital stay, and lower total hospitalization costs. Early surgery was associated with fewer postoperative complications, consistent with a previous study [8]. Elderly patients are vulnerable and the prevention of postoperative complications may be associated with favorable functional outcomes and improved quality of life [26]. Similar to the case for geriatric hip fracture patients, this may have arisen because early surgery in geriatric distal femur fracture patients enabled the early initiation of rehabilitation, increasing the chance for better functional recovery, resulting in fewer postoperative complications, and having positive impacts on hospital stay and total hospitalization costs [18,27]. The mechanism for the benefits of early surgery on functional outcomes should be addressed in future studies.

The present study had several strengths. To the best of our knowledge, this was the first large-scale study to demonstrate the effectiveness of early surgery for distal femur fracture in elderly patients on in-hospital mortality, postoperative complications, length of hospital stay, and total hospitalization costs. The study was able to compensate for limitations of previous studies that arose from their small sample sizes and low external validity due to the use of single-center data. Targeting the performance of surgery within two days of admission represents a significant change in practice, because 85.7% of the patients in the study did not receive surgery within two days. The early timing of surgery in hip fracture is already recognized worldwide as a quality indicator for the assessment of hospital performance; therefore, the results of the present study may inform existing distal femur fracture care guidelines and policies.

This study also has some limitations. First, this study may have immortal time biases for the two time points—namely, the time from hospital admission to surgery and time after early discharge [28]. In this study, time from hospital admission to surgery was considered immortal because the performance of surgery implied that the patients survived until surgery. Therefore, the delayed surgery group had a guaranteed survival advantage over the early surgery group because of the immortal time from hospital admission to surgery. Meanwhile, time after early discharge was considered immortal because patients who were discharged alive were considered to remain alive at 30 days in this study. Therefore, the early surgery group had a guaranteed survival advantage over the delayed surgery group because of the immortal time after early discharge. Second, this observational study using a real-world database unmeasured confounding variables. Thus, individual surgeons decided the timing of surgery according to the criteria in their own settings, which would lead to confounding by indication. We attempted to control for possible confounding factors described in previous reports, including covariates of patient and surgeon-hospital characteristics. However, we were unable to obtain detailed data on the distal femur fracture type and the time from injury to admission. We assumed that the day of injury was the same as the day of hospital admission. The appropriate timing for surgery needs to be further investigated with better time resolution, such as hours, instead of number of days from admission to surgery in future studies. Third, the Japanese Diagnosis Procedure Combination inpatient database does not contain links to data after discharge. Therefore, we could not evaluate the outcomes after longer follow-up (90 or 365 days). Previous studies showed that delayed surgery by more than two days in geriatric distal femur fracture patients was significantly associated with increased 1-year mortality [7,8]. In addition, the assumption that patients who were discharged alive within 30 days of hospital admission remained alive at 30 days could lead to misclassification for 30-day in-hospital mortality. Therefore, further studies are needed to examine the effectiveness of early surgery on mortality with longer follow-up periods. Fourth, there is a lack of external validity because all the data were obtained from a Japanese database. It remains unclear whether the results of the study can be generalized to other countries with different patient characteristics and healthcare systems.

## 5. Conclusions

This nationwide observational study suggested that early surgery within two days of hospital admission was not associated with a reduction in 30-day mortality in patients with geriatric distal femur fracture. However, early surgery was associated with decreased postoperative complications and lower total hospitalization costs.

## Figures and Tables

**Figure 1 jcm-10-05800-f001:**
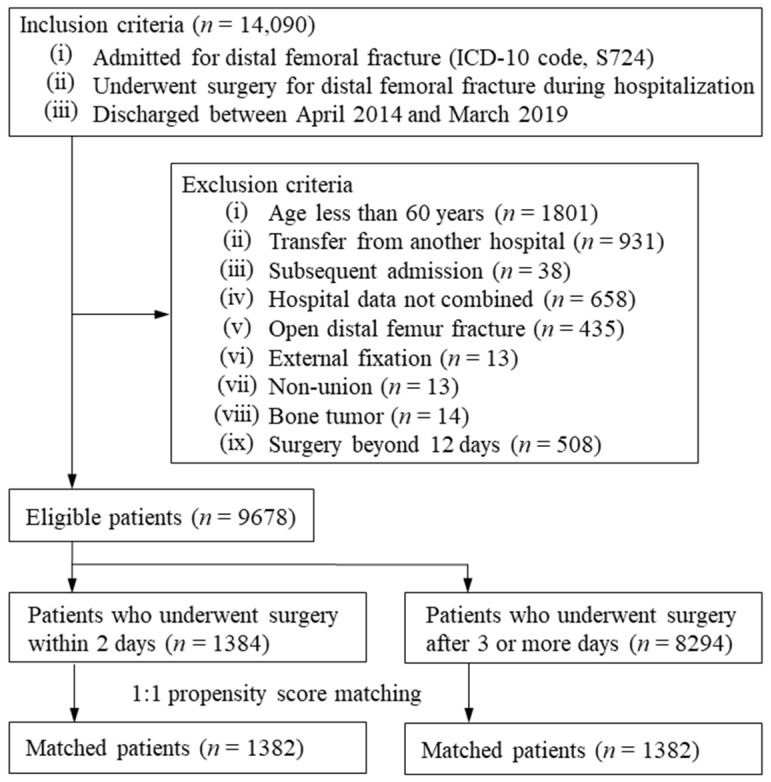
Flow chart for patient inclusion and exclusion.

**Figure 2 jcm-10-05800-f002:**
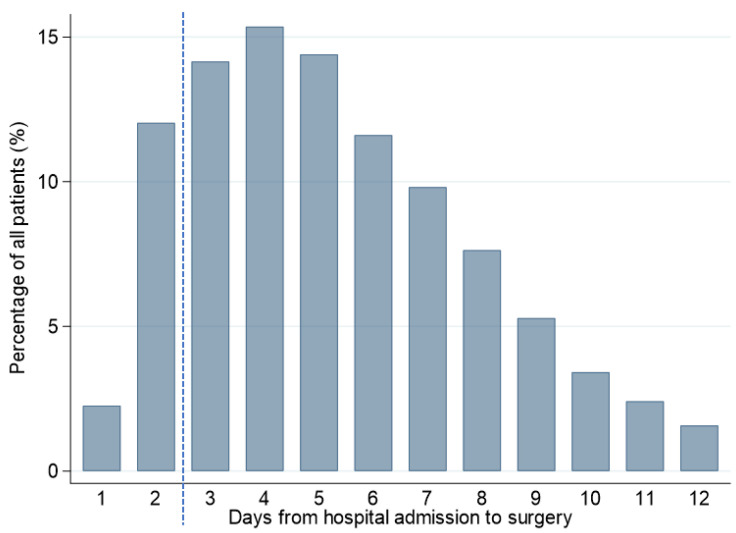
Days from hospital admission to surgery. The dotted line shows the border between the early and delayed surgery groups.

**Table 1 jcm-10-05800-t001:** Patient characteristics before and after propensity score matching.

	Unmatched Cohort			Matched Cohort		
	Early Surgery (≤2 Days) *n* = 1384	Delayed Surgery (≥3 Days) *n* = 8294	ASD	Early Surgery (≤2 Days) *n* = 1382	Delayed Surgery (≥3 Days) *n* = 1382	ASD
Age (years), mean (SD)	81.3 (9.4)	81.1 (9.3)	2.3	81.3 (9.4)	81.4 (9.5)	0.7
Female sex, *n* (%)	1227 (88.7)	7515 (90.6)	6.4	1225 (88.6)	1214 (87.8)	2.6
Body mass index (kg/m^2^), *n* (%)						
<18.5	264 (19.1)	1495 (18.0)	2.7	264 (19.1)	262 (19.0)	0.4
18.5–24.9	718 (51.9)	4384 (52.9)	2	716 (51.8)	701 (50.7)	2.2
25.0–29.9	235 (17.0)	1478 (17.8)	2.2	235 (17.0)	252 (18.2)	3.2
≥30	64 (4.6)	399 (4.8)	0.9	64 (4.6)	69 (5.0)	1.7
Missing	103 (7.4)	538 (6.5)	3.8	103 (7.5)	98 (7.1)	1.4
Smoking history, *n* (%)						
Non-smoker	1189 (85.9)	7080 (85.4)	1.6	1187 (85.9)	1182 (85.5)	1.0
Current/past smoker	96 (6.9)	602 (7.3)	1.3	96 (6.9)	99 (7.2)	0.8
Missing	99 (7.2)	612 (7.4)	0.9	99 (7.2)	101 (7.3)	0.6
Unconscious at admission, *n* (%)	186 (13.4)	979 (11.8)	4.9	184 (13.3)	180 (13.0)	0.9
Home medical care use, *n* (%)	107 (7.7)	693 (8.4)	2.3	107 (7.7)	106 (7.7)	0.3
Admission from nursing home, *n* (%)	246 (17.8)	1281 (15.4)	6.3	245 (17.7)	232 (16.8)	2.5
Ambulance use, *n* (%)	796 (57.5)	5129 (61.8)	8.8	795 (57.5)	794 (57.5)	0.1
Admission on weekend, *n* (%)	182 (13.2)	3444 (41.5)	67.1	182 (13.2)	183 (13.2)	0.2
Calendar year, *n* (%)						
2014	282 (20.4)	1862 (22.4)	5.1	282 (20.4)	274 (19.8)	1.4
2015	284 (20.5)	1750 (21.1)	1.4	284 (20.5)	311 (22.5)	4.8
2016	280 (20.2)	1633 (19.7)	1.4	280 (20.3)	250 (18.1)	5.5
2017	278 (20.1)	1595 (19.2)	2.2	276 (20.0)	291 (21.1)	2.7
2018	260 (18.8)	1454 (17.5)	3.3	260 (18.8)	256 (18.5)	0.7
Comorbidities, *n* (%)						
Dementia						
Absent	861 (62.2)	5239 (63.2)	2	860 (62.2)	855 (61.9)	0.7
Mild	261 (18.9)	1702 (20.5)	4.2	260 (18.8)	273 (19.8)	2.4
Moderate/severe	262 (18.9)	1353 (16.3)	6.9	262 (19.0)	254 (18.4)	1.5
Myocardial infarction	4 (0.3)	80 (1.0)	8.6	4 (0.3)	5 (0.4)	0.9
Chronic heart failure	69 (5.0)	609 (7.3)	9.8	69 (5.0)	68 (4.9)	0.3
Peripheral vascular disease	11 (0.8)	98 (1.2)	3.9	11 (0.8)	14 (1.0)	2.2
Cerebrovascular disease	120 (8.7)	783 (9.4)	2.7	120 (8.7)	115 (8.3)	1.3
Chronic pulmonary disease	39 (2.8)	286 (3.4)	3.6	39 (2.8)	38 (2.7)	0.4
Rheumatic disease	44 (3.2)	436 (5.3)	10.3	44 (3.2)	37 (2.7)	2.5
Peptic ulcer	40 (2.9)	264 (3.2)	1.7	40 (2.9)	41 (3.0)	0.4
Mild liver dysfunction	46 (3.3)	389 (4.7)	7	46 (3.3)	40 (2.9)	2.2
Diabetes mellitus without complications	225 (16.3)	1571 (18.9)	7.1	224 (16.2)	197 (14.3)	5.1
Diabetes mellitus with complications	21 (1.5)	239 (2.9)	9.3	21 (1.5)	23 (1.7)	1
Hemiplegia	14 (1.0)	57 (0.7)	3.5	14 (1.0)	15 (1.1)	0.8
Renal dysfunction	28 (2.0)	307 (3.7)	10.1	28 (2.0)	41 (3.0)	5.6
Malignancy	37 (2.7)	277 (3.3)	3.9	37 (2.7)	35 (2.5)	0.8
Severe liver dysfunction	3 (0.2)	9 (0.1)	2.7	3 (0.2)	4 (0.3)	1.8
Charlson comorbidity index	0.7 (1.0)	0.9 (1.1)	17.9	0.7 (1.0)	0.7 (0.9)	1.1
Trauma mortality prediction score, mean (SD)	3.6 (1.5)	3.5 (1.5)	4.5	3.6 (1.5)	3.6 (1.7)	1.1
ICU/HCU at admission, *n* (%)	59 (4.3)	195 (2.4)	10.7	57 (4.1)	56 (4.1)	0.4
Periprosthetic fracture, *n* (%)	92 (6.6)	672 (8.1)	5.6	92 (6.7)	94 (6.8)	0.6
Operation, *n* (%)						
Treatment with plating	772 (55.8)	4813 (58.0)	4.5	772 (55.9)	799 (57.8)	3.9
Treatment with nailing	462 (33.4)	2684 (32.4)	2.2	460 (33.3)	429 (31.0)	4.8
Treatment with arthroplasty	9 (0.7)	36 (0.4)	2.9	9 (0.7)	6 (0.4)	3
Treatment unknown	141 (10.2)	761 (9.2)	3.4	141 (10.2)	148 (10.7)	1.7
Academic hospital, *n* (%)	219 (15.8)	1206 (14.5)	3.6	219 (15.8)	216 (15.6)	0.6
Teaching hospital, *n* (%)	1064 (76.9)	6298 (75.9)	2.2	1062 (76.8)	1047 (75.8)	2.6
Tertiary hospital, *n* (%)	447 (32.3)	2226 (26.8)	12	445 (32.2)	457 (33.1)	1.9
Hospital beds, *n* (%)						
<200	196 (14.2)	1307 (15.8)	4.5	196 (14.2)	202 (14.6)	1.2
200–399	528 (38.2)	3335 (40.2)	4.2	527 (38.1)	532 (38.5)	0.7
400–599	463 (33.5)	2398 (28.9)	9.8	462 (33.4)	458 (33.1)	0.6
600–799	160 (11.6)	900 (10.9)	2.2	160 (11.6)	152 (11.0)	1.8
>800	37 (2.7)	354 (4.3)	8.7	37 (2.7)	38 (2.7)	0.4
Hospital volume, *n* (%)	5.0 (2.4)	4.5 (2.4)	19.1	5.0 (2.4)	5.0 (2.6)	1.0

ASD, absolute standardized difference; ICU, intensive care unit; HCU, high care unit. Body mass index was calculated as weight in kilograms divided by height in meters squared.

**Table 2 jcm-10-05800-t002:** Outcomes in the original unmatched cohort and matched cohort.

	Unmatched Cohort		Matched Cohort			
	Early Surgery (≤2 Days) *n* = 1384	Delayed Surgery (≥3 Days) *n* = 8294	Early Surgery (≤2 Days) *n* = 1382	Delayed Surgery (≥3 Days) *n* = 1382	Risk Difference (95% CI)	*p*-Value
Primary outcome						
30-day mortality, *n* (%)	12 (0.9)	41 (0.5)	12 (0.9)	12 (0.9)	0.0 (−0.7 to 0.7)	>0.999
Secondary outcomes						
In-hospital mortality, *n* (%)	21 (1.5)	106 (1.3)	21 (1.5)	21 (1.5)	0.0 (−0.9 to 0.9)	>0.999
Composite outcome, *n* (%)	187 (13.5)	1383 (16.7)	186 (13.5)	237 (17.1)	−3.7 (−6.5 to −0.9)	0.007
Length of hospital stay (days), mean (SD)	42 (42)	53 (36)	42 (42)	50 (36)	−8.4 (−11.8 to −5.0)	<0.001
Length of time from surgery to discharge (days), mean (SD)	41 (42)	48 (36)	41 (42)	45 (36)	−4.5 (−7.9 to −1.0)	0.011
In-hospital costs (US dollars), mean (SD)	16,099 (10,721)	18,844 (10,803)	16,101 (10,729)	18,202 (9859)	−2101 (−2991 to −1212)	<0.001

Hospital costs were calculated using the exchange rate of 1 US dollar = 110 Japanese yen.

**Table 3 jcm-10-05800-t003:** Subgroup analyses for 30-day mortality in the propensity score-matched cohort.

	Early Surgery	Delayed Surgery	Risk Difference (95% CI)	*p*-Value
Admission on weekend				
30-day mortality, *n* (%)	1/182 (0.5)	2/183 (1.1)	−0.5 (−2.4 to 1.3)	0.571
Charlson comorbidity index ≥ 1				
30-day mortality, *n* (%)	4/652 (0.6)	6/670 (0.9)	−0.3 (−1.2 to 0.7)	0.552
Hospital size < 400 beds				
30-day mortality, *n* (%)	5/723 (0.7)	8/734 (1.1)	−0.4 (−1.4 to 0.6)	0.415
Hospital volume < 4				
30-day mortality, *n* (%)	5/548 (0.9)	5/578 (0.9)	−0.0 (−1.1 to 1.2)	0.933

## Data Availability

The datasets analyzed during the current study are not publicly available owing to contracts with the hospitals providing data to the database.

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
