# Peer review of "Associations between Early Surgery and Postoperative Outcomes in Elderly Patients with Distal Femur Fracture: A Retrospective Cohort Study"

_jcm, 2021, doi:10.3390/jcm10245800_

Round 1

Reviewer 1 Report

Thank you for the opportunity to review this manuscript. The authors have generated an analysis of 5 years of a national database to evaluate the outcomes of elderly distal femur fractures. They have compared early and delayed surgical treatment. The study is interesting, topical and presents a 10,000 foot view of the problem with a large cohort of patients. 

i commend the authors for their work here. I do think this is a topical question and one with currently conflicting results in the literature. The current study does add to the available data however ther are a few areas where the study can be strengthened or the man’s culprit should be made clearer.

1) While theoretically true that this is a retrospective cohort study the authors should more accurately describe this as a big data study given they had no direct input on the cohort designs or data itself. 

2) Line 84- can the authors clarify what this sentence means? Where else are these data recorded? Please clarify. If they are not in the database why not? And where did the authors get this data then?

3) Line 108 - timing TO surgery after admission. 

4) Line 116 - I strongly feel the authors should reconsider their description of their primary outcome. 30-day mortality was  inferred here as the database does not capture all data (line 116-117) beyond discharge. Potentially re-naming this to in-hospital mortality… or some other factor. Can the authors discuss how this all-cause-in hospital mortality ( a secondary outcome) is different? 

5) Was a power calculation performed?

6) Line 184 - can the authors synchronize how the early surgery goop appeared to have more patients in the ICU/High care unit? This would seem at odds with an early surgical cohort?

7)Authors should comment on the group differences here? Once could argue that given that close to 50% (41.5%) of their delayed cases were admitted to the weekend vs. 13% in the early group that potentially the sicker patients admitted during the weekday were potentially ‘offset’ by the healthy patients admitted on a weekend. While propensity matching may accommodate some of these changes it will not factor into additional ‘behind the scenes’ delayed factors. 

8)Line - 256 - can the authors discuss that 30 day mortality vs. in hospital mortality vs. their composite outcome. I worry that given they have ‘assumed’ those making it to discharge pre-e-days all ‘survived’ this may artifactual improve the 30 day mortality rate This is particularly true if early surgical patients were more likely to be discharged early. Hence we could be seeing a sampling bias.

9) An additional metric that may be interesting would be days post-op to discharge. This may more appropriately evaluate for additional complications/complexities if the early surgical patients go home POD#2 Vs. delayed surgical cases that go home POD#5 etc.  Suggest including. 

10) Line 268 - I would caution the authors about speculating about the effects of timing of surgery and rehab or mobilization given that the current study does not examine these factors with any granularity beyond days of hospitalization.

11)Line 275 - The authors should remove short term morality here as they have not specifically shown short term mortality differences unless referring specifically to in-hospital mortality?

12)Line 284 - I commend the authors for commenting on immortality bias - I wonder however if the reverse is not true and actually if there is a bias towards staying in hospital. Currently if subjects remain in hospital and die their mortality is counted as a death while patients discharged day 13 post op and who die at home or in a facility will not be ‘captured’ by the database as a. Mortality and thus skew the results for survival. 

Overall this is an interesting paper. I think there are a number of clarifications that need to be made in terms of terminology and clarifications of methodology however the manuscript given these limitations provides some additional information on an important topic. 

Author Response

Response to Reviewer 1 Comments:

Thank you for the opportunity to review this manuscript. The authors have generated an analysis of 5 years of a national database to evaluate the outcomes of elderly distal femur fractures. They have compared early and delayed surgical treatment. The study is interesting, topical and presents a 10,000 foot view of the problem with a large cohort of patients. 

Response: We thank Reviewer 1 very much for these comments.

Point 1: i commend the authors for their work here. I do think this is a topical question and one with currently conflicting results in the literature. The current study does add to the available data however there are a few areas where the study can be strengthened or the man’s culprit should be made clearer.

Response 1: As the reviewer correctly pointed out, our study has some limitations. We have added the sentences shown below as a further limitation of the study.

Line 304: “Fourth, there is a lack of external validity because all of the data were obtained from a Japanese database. It remains unclear whether the results of the study can be generalized to other countries with different patient characteristics and healthcare systems.”

Point 2: 1) While theoretically true that this is a retrospective cohort study the authors should more accurately describe this as a big data study given they had no direct input on the cohort designs or data itself. 

Response 2: We have revised the related sentences as shown below.

Line 67: “This study was a retrospective cohort study using a national administrative inpatient database”

Line 76: “We used the Japanese Diagnosis Procedure Combination inpatient database under the management of the Ministry of Health, Labour, and Welfare,”

Point 3: 2) Line 84- can the authors clarify what this sentence means? Where else are these data recorded? Please clarify. If they are not in the database why not? And where did the authors get this data then?

Response 3: The indicated data were recorded in the same database. We have revised the related sentence as shown below.

Line 84: “The diagnoses at admission, comorbidities at admission, and complications during hospitalization are recorded in the database.”

Point 4: 3) Line 108 - timing TO surgery after admission. 

Response 4: We have changed the indicated description in accordance with the reviewer’s suggestion.

Point 5: 4) Line 116 - I strongly feel the authors should reconsider their description of their primary outcome. 30-day mortality was inferred here as the database does not capture all data (line 116-117) beyond discharge. Potentially re-naming this to in-hospital mortality… or some other factor. Can the authors discuss how this all-cause-in hospital mortality ( a secondary outcome) is different? 

Response 5: We agree with the reviewer’s comments. As the reviewer indicated, “30-day in-hospital mortality” is the correct description. We have revised the manuscript accordingly.

Point 6: 5) Was a power calculation performed?

Response 6: We thank the reviewer for this comment. Because the study was a retrospective database study, prespecified sample size calculations were not performed. All patients with available data were considered eligible for the study.

Point 7: 6) Line 184 - can the authors synchronize how the early surgery group appeared to have more patients in the ICU/High care unit? This would seem at odds with an early surgical cohort?

Response 7: As the reviewer correctly pointed out, there was a difference in ICU/HCU admission between the two groups in the unmatched cohort. This indicates that the patients in the early surgery group may have included a greater proportion of critically ill patients than the patients in the control group. Propensity score methods can help to adjust for this type of selection bias [Ann Intern Med. 2010;152(6):393–395; Am Stat. 1985;39(1):33]. These methods attempt to construct a randomized trial-like situation wherein the groups become comparable for the observed prognostic characteristics. Our study ensured that the patient characteristics were well-balanced between the two groups in the weighted cohort by calculating the absolute standardized differences (less than 10%) (Table 1). Therefore, the patients in the early and delayed surgery groups were comparable for the observed prognostic characteristics.

Point 8: 7)Authors should comment on the group differences here? Once could argue that given that close to 50% (41.5%) of their delayed cases were admitted to the weekend vs. 13% in the early group that potentially the sicker patients admitted during the weekday were potentially ‘offset’ by the healthy patients admitted on a weekend. While propensity matching may accommodate some of these changes it will not factor into additional ‘behind the scenes’ delayed factors. 

Response 8: In the well-balanced groups after propensity score matching, the patients in the early and delayed surgery groups were comparable for the observed prognostic characteristics such as admission on weekend or sickness situation at admission. Please see our Response 7.

As the reviewer pointed out, propensity score methods cannot adjust for unmeasured confounding variables. We have revised the related sentence as shown below.

Line 289: “Second, this observational study using a real-world database has unmeasured confounding variables.”

Point 9: 8)Line - 256 - can the authors discuss that 30 day mortality vs. in hospital mortality vs. their composite outcome. I worry that given they have ‘assumed’ those making it to discharge pre-e-days all ‘survived’ this may artifactual improve the 30 day mortality rate This is particularly true if early surgical patients were more likely to be discharged early. Hence we could be seeing a sampling bias.

Response 9: We agree with the reviewer’s comments. We have added a related sentence to the limitations of the study as shown below.

Line 303: “In addition, the assumption that patients who were discharged alive within 30 days of hospital admission remained alive at 30 days could lead to misclassification for 30-day in-hospital mortality.”

Point 10: 9) An additional metric that may be interesting would be days post-op to discharge. This may more appropriately evaluate for additional complications/complexities if the early surgical patients go home POD#2 Vs. delayed surgical cases that go home POD#5 etc.  Suggest including. 

Response 10: We thank the reviewer for this suggestion. Length of time from surgery to discharge is an interesting outcome because patients with postoperative complications would probably require more days from surgery to discharge than patients without complications. Therefore, we have added length of time from surgery to discharge as a secondary outcome, and revised Table 2 accordingly. We have revised the related sentences as shown below.

Line 118: “The secondary outcomes were all-cause in-hospital mortality, composite outcome of death or postoperative complications during hospitalization, length of hospital stay, length of time from surgery to discharge, and total hospitalization costs [19].”

Line 230: “Patients in the early surgery group had significantly shorter length of hospital stay (risk difference, −8.4 days; 95% CI, −11.8 to −5.0 days), shorter length of time from surgery to discharge (risk difference, −4.5 days; 95% CI, −7.9 to −1.0 days), and lower total hospitalization costs (risk difference, −2,101 US dollars; 95% CI, −2,991 to −1,212 US dollars) than patients in the delayed surgery group.”

Point 11: 10) Line 268 - I would caution the authors about speculating about the effects of timing of surgery and rehab or mobilization given that the current study does not examine these factors with any granularity beyond days of hospitalization.

Response 11: As the reviewer pointed out, it is a possible mechanism based on our results. We have revised the related sentences as shown below.

Line 268: “Similar to the case for geriatric hip fracture patients, this may have arisen because early surgery in geriatric distal femur fracture patients enabled early initiation of rehabilitation, increasing the chance for better functional recovery, resulting in fewer postoperative complications, and having positive impacts on hospital stay and total hospitalization costs [18,28]. The mechanism for the benefits of early surgery on the functional outcomes should be addressed in future studies.”

Point 12: 11)Line 275 - The authors should remove short term morality here as they have not specifically shown short term mortality differences unless referring specifically to in-hospital mortality?

Response 12: We have revised the manuscript in accordance with the reviewer’s suggestion.

Point 13: 12)Line 284 - I commend the authors for commenting on immortality bias - I wonder however if the reverse is not true and actually if there is a bias towards staying in hospital. Currently if subjects remain in hospital and die their mortality is counted as a death while patients discharged day 13 post op and who die at home or in a facility will not be ‘captured’ by the database as a. Mortality nd thus skew the results for survival. 

Response 13: As the reviewer pointed out, an immortal time bias may exist depending on the time point. In this study, the time points were surgery and discharge within 30 days of hospital admission. We have revised the related sentences as shown below.

Line 284: “First, the study may have immortal time biases for the two time points, namely time from hospital admission to surgery and time after early discharge [28]. In the study, time from hospital admission to surgery was considered immortal because performance of surgery implied that the patients survived until surgery. Therefore, the delayed surgery group had a guaranteed survival advantage over the early surgery group because of the immortal time from hospital admission to surgery. Meanwhile, time after early discharge was considered immortal because patients who were discharged alive were considered to remain alive at 30 days in the study. Therefore, the early surgery group had a guaranteed survival advantage over the delayed surgery group because of the immortal time after early discharge.”

Reviewer 2 Report

This is an interesting study answering an everyday life pragmatic question about when to treat older patients affected by distal femur fracture. Authors elegantly showed that although mortality at 30 days was not significantly different in patients treated early or late, this impacted on post-operative complications and hospital costs. Moreover, as hospital stays in Japan are traditionally longer than in other Western countries, this increases the significance of data due to the strict monitoring of patients' status both in the short and the long term. 

  • Keywords should be revised: "distal femur fracture" and "distal femoral fracture" are almost the same; additional pertinent kewyord may be added in order to improve indexing and retrieval.

Author Response

Response to Reviewer 2 Comments:

This is an interesting study answering an everyday life pragmatic question about when to treat older patients affected by distal femur fracture. Authors elegantly showed that although mortality at 30 days was not significantly different in patients treated early or late, this impacted on post-operative complications and hospital costs. Moreover, as hospital stays in Japan are traditionally longer than in other Western countries, this increases the significance of data due to the strict monitoring of patients' status both in the short and the long term. 

Response: We thank Reviewer 2 very much for these comments.

Point 1: Keywords should be revised: "distal femur fracture" and "distal femoral fracture" are almost the same; additional pertinent keyword may be added in order to improve indexing and retrieval.

Response 1: We have revised the keywords as shown below.

“Keywords: distal femur fracture; surgical timing; mortality; complications; length of hospital stay; medical costs; database”